# Viral dUTPases: Modulators of Innate Immunity

**DOI:** 10.3390/biom12020227

**Published:** 2022-01-28

**Authors:** Maria Eugenia Ariza, Brandon Cox, Britney Martinez, Irene Mena-Palomo, Gloria Jeronimo Zarate, Marshall Vance Williams

**Affiliations:** 1Department of Cancer Biology and Genetics, The Ohio State University Wexner Medical Center, Columbus, OH 43210, USA; Maria.Ariza@osumc.edu; 2Institute for Behavioral Medicine Research, The Ohio State University Wexner Medical Center, Columbus, OH 43210, USA; Brandon.Cox@osumc.edu (B.C.); Britney.Martinez@osumc.edu (B.M.); Irene.MenaPalomo@osumc.edu (I.M.-P.); gloriajz881@gmail.com (G.J.Z.)

**Keywords:** deoxyuridine triphosphate nucleotidohydrolase, dUTPase, double-stranded DNA viruses

## Abstract

Most free-living organisms encode for a deoxyuridine triphosphate nucleotidohydrolase (dUTPase; EC 3.6.1.23). dUTPases represent a family of metalloenzymes that catalyze the hydrolysis of dUTP to dUMP and pyrophosphate, preventing dUTP from being incorporated into DNA by DNA polymerases, maintaining a low dUTP/dTTP pool ratio and providing a necessary precursor for dTTP biosynthesis. Thus, dUTPases are involved in maintaining genomic integrity by preventing the uracilation of DNA. Many DNA-containing viruses, which infect mammals also encode for a dUTPase. This review will summarize studies demonstrating that, in addition to their classical enzymatic activity, some dUTPases possess novel functions that modulate the host innate immune response.

## 1. Introduction

While there are some exceptions, most free-living organisms encode for a deoxyuridine triphosphate nucleotidohydrolase (dUTPase; EC 3.6.1.23) [1]. In addition, many viruses which infect archaea, bacteria, plants and animals also encode for a dUTPase. Studies have demonstrated that the genes encoding for the dUTPases are essential in bacteria [2,3], with the exception of some species [1], yeast [4] and mice [5], suggesting that this enzyme is required for survival/life. This concept is supported by the extensive number of species that possess dUTPases. Pfam, a database of protein families generated on annotations and sequence alignments using hidden Markov models, has demonstrated the presence of putative dUTPases in at least 8358 species [6]. Data used in this manuscript were collected primarily by Pubmed and pBLAST searches.

dUTPases can be divided into at least three structurally distinct families based upon an ordered series of conserved motifs (N-terminal to C-terminal; I, II, III, IV, and V) and structure. The largest family, which exhibits a high specificity for dUTP, is the homotrimeric dUTPases, found in plants, animals, fungi, bacteria and some viruses, including some adenoviruses, poxviruses and retroviruses. These homotrimeric dUTPases, of which prototypes include *Escherichia coli* and human, are composed of three identical polypeptides, each folded in an eight-stranded jelly-roll beta barrel and aligned so that the three highly conserved amino acid motifs I, II and IV on one polypeptide interact with the amino acids in conserved motif III of an adjacent peptide. The catalytic site is completed with the binding of dUTP to the flexible C-terminal end of the third polypeptide, which contains the conserved amino acids in motif V. Thus, all three polypeptide subunits contribute to the formation of a catalytic site and there are three catalytic sites per holoenzyme [7,8].

The homodimeric dUTPases, identified in *Leishmania major* [9,10], *Trypanosoma cruzi* [11] and *Campylobacter jejuni* [12], exist as α-helical dimeric proteins and exhibit broader substrate specificity, hydrolyzing both dUDP and dUTP. Furthermore, sequence comparisons revealed that the dimeric dUTPases lack the five consensus amino acid motifs found in mono and trimeric dUTPases and they are evolutionary related to the dCTPase-dUTPase of bacteriophages T2 and T4 [11].

The monomeric dUTPases are found exclusively in some members of the herpesviruses [13]. The herpesvirus dUTPases are approximately two-fold larger than the human and *E. coli* dUTPase homotrimers and the conserved motif III has been displaced toward the amino terminal end of the molecule. Thus, in monomeric dUTPases, the conserved motif arrangement is III, I II, IV and V when compared to the arrangement observed in homotrimeric dUTPases. Structural data on the Epstein–Barr virus (EBV)-encoded dUTPase have demonstrated that the single catalytic site of the enzyme is comprised of the five highly conserved motifs characteristic of the homotrimeric dUTPases [14]. 

Despite advances in understanding the evolutionary history of organisms made possible by molecular phylogenetics, the origins of most viruses and their divergence during evolution remain very poorly understood. This is due to the enormous diversity of virus types. Classification of viruses is usually based on the morphology of the virus particle and the composition of the genome as defined by the International Committee of Virus Taxonomy. However, other approaches to virus phylogeny are the use of the three-dimensional structure of viral capsid proteins [15] and the linkage of virus evolution to polintons, large DNA transposons, containing genes with homology to some virus genes, using protein and genome structures [16,17].

The evolutionary origin of the viral dUTPase gene (*dUT*) is obscure (Figure 1). Several studies have addressed the possible mechanisms by which DNA viruses acquire this gene. These studies utilized bioinformatic approaches and various model systems to compare the amino acid sequence relationships between eukaryotes, eubacteria and archaeal organisms as well as viruses [18,19,20,21]. The most comprehensive study included the analysis of 24 host and 50 viral dUTPase sequences [18,19]. These studies provided evidence that at least five horizontal acquisitions of viral *dUT* genes have occurred from hosts. It has been proposed that since the *dUT* genes in DNA viruses that infect eukaryotic organisms lack introns, they must have arisen from a cDNA of an mRNA in a mechanism analogous to that occurring in the acquisition of some viral oncogenes [18]. Caprari et al. [22] suggested that the dUTPases of double-stranded DNA viruses and retroviruses were obtained through different evolutionary events, but were acquired to serve the same function in different viral groups. However, this assumption is based upon the premise that the only function of these dUTPase proteins is to enzymatically maintain genomic integrity. However, could these dUTPase proteins have additional unknown functions that are also important for viral replication/maintenance?

The monomeric dUTPases which exhibit an altered order series of conserved motifs (N-terminal to C-terminal; III, I, II, IV, and V) are reported to have evolved through a gene duplication event(s). This family of dUTPases are only observed in the Alpha- and Gammaherpesvirinae subfamilies of the *Herpesvirales* [18,20]. While members of the *Betaherpesvirinae* subfamily contain a gene corresponding to the genomic location of *dUT* genes in the Alpha- and Gammaherpesvirinae subfamilies, these genes lack the conserved motifs defining dUTPases; and the proteins encoded by these regions lack apparent enzymatic activity [23,24,25]. Sequence analyses have demonstrated that a motif referred to as motif VI, which is located between motifs II and IV within the monomeric dUTPases, is conserved in Alpha-, Beta- and Gammaherpesvirinae, but is absent in the homotrimeric dUTPases [20] (Figure 2). McGeehan et al. [20] were the first to suggest that the monomeric dUTPases might have additional functions not related to the enzymatic activity. However, until recently, this has not been explored.

Most studies have focused on biochemical characterization of the enzymatic activity and structural characteristics of dUTPases from various organisms. Studies have demonstrated that some viruses which encode for dUTPases require the enzyme for replication in terminally differentiated cells in which cellular dUTPase activity is low or absent. This probably reflects the importance of the dUTPase in these viruses for maintaining genomic integrity. Recent studies also demonstrated that some dimeric and trimeric dUTPases encoded by bacteriophages regulate the transfer of mobile genetic elements in various strains of *Staphylococcus aureus* thus possibly contributing to their pathogenicity [26,27,28]. Furthermore, there is accumulating evidence that virus-encoded dUTPases may have undiscovered roles in several cellular processes and responses that can contribute not only to disease but to aging as well [29,30]. The purpose of this review is to highlight the recent advances and to highlight the critical need for conducting further research on these proteins to elucidate their role in human disease. 

## 2. dUTPases of Double-Stranded DNA Viruses: DNA Sequencing and Annotation

With the exception of members of the *Ascoviridae* family, which contain four virus species that infect moths, all double-stranded DNA virus families contain members that possess putative dUTPase-encoding genes (Table 1). Except for members of the Alpha- and Gammaherpesvinirae subfamilies, all these putative dUT genes encode for proteins that exhibit an ordered series of conserved motifs (N-terminal to C-terminal; I, II, III, IV, and V), which suggests a homotrimeric structure. The smallest dUTPases that have been shown to be enzymatically functional are the homotrimeric FIV dUTPase (133 amino acids) and the monomeric EBV dUTPase (278 amino acids) (Table 2). The predicted amino acid length of dUTPase proteins determined by annotation and from experimental data range from 142 to 478 amino acids, suggesting that other non-identified conserved motifs/domains may exist, which might have additional unknown functions of these proteins.

### 2.1. Baculoviridae

Members of the family Baculoviridae are rod-shaped viruses with circular, covalently closed, double-stranded DNA genomes. This family includes four genera: Alphabaculovirus (lepidopteran-specific nucleopolyhedroviruses), Betabaculovirus (lepidopteran-specific granuloviruses), Gammabaculovirus (hymenopteran-specific) and Deltabaculovirus (dipteran-specific). While putative *dUT* genes have been identified in some members of this family by DNA sequencing and annotation, there have not been any studies to characterize the gene products either biochemically or biophysically. No putative *dUT* genes have been identified in members of the Delta- or Gammabaculovirus genera, suggesting that these genes might have been lost [31,32,33].

Phylogenetic analysis of the *dUT* gene in *Baculoviridae* suggests that the gene was dispersed in this family of viruses, requiring at least nine horizontal gene transfer events from several sources including other baculoviruses [31]. Of interest was a genomic study on the *pelu112* gene of *Perigonia lusca* single nucleopolyhedrovirus (PeluSNPV), a group II alphabaculovirus. This gene was predicted to encode a fusion of thymidylate kinase (*tmk*) and dUTPase (*dut*). The *pelu112* gene encodes for a protein of 317 amino acids and the C-terminal 152 amino acids exhibits homology to the five conserved motifs characteristic of trimeric dUTPases. Furthermore, *pelu112* has homologs in two distantly related baculoviruses, *Erinnyis ello granulovirus* (ORF-5; 317 amino acids), a beta baculovirus and *Orgyia pseudotsugata* multiple nucleopolyhedrovirus (ORF31; 317 amino acids), an alpha baculovirus [31].

### 2.2. Giant Viruses

Viruses are classified as ‘giant’ based on the size of their virion and genome coding capacity. These nucleocytoplasmic large double-stranded DNA viruses (NCLDV) have genomes in excess of 2.0 × 10^6^ base pairs encoding for more than 500 proteins, rivaling in sizes and gene contents those of cellular microbes. The superfamily of NCLDV also includes *Iridoviridae*, *Poxviridae*, *Asfarviridae* and *Ascoviridae*. The *dUT* gene is one of 33 polyphyletic conserved genes identified in NCLDV [34].

## 3. dUTPases of Double-Stranded DNA Viruses and Retroviruses: Functional Enzymatic Activity

### 3.1. Asfarviridae: Asfivirus

African swine fever virus (ASFV) is a large enveloped double-stranded DNA virus that causes a deadly infection in pigs, approaching 100% mortality in domestic pigs. Studies on the ASFV dUTPase have focused on its catalytic properties [35] and structure [36,37,38]. While the ASFV dUTPase is a homotrimer, the protein has three active sites, all built by two adjacent chains [38]. A rather unique feature is that it displays a non-canonical folding pattern that differs from that of the classic homotrimeric dUTPases in that the active site is composed of two subunits. 

### 3.2. Iridoviridae

*Iridoviridae* comprise a collection of large icosahedral, double-stranded DNA-containing viruses that are classified into two subfamilies: *Alphairidovirinae*, whose members infect primarily vertebrates such as bony fish, amphibians and reptiles, and *Betairidovirinae*, whose members infect mainly invertebrates such as insects and crustaceans. Only a single protein encoded gene (*dUT*) from *Rana grylio* (RGV), a pig frog pathogen, has been examined in any detail. The 164 amino acid recombinant protein encodes for an enzymatically functional dUTPase [39]. Furthermore, sequence comparisons and multiple alignments suggested that the RGV dUTPase functions as a homotrimer. 

### 3.3. Nimaviridae: Whispovirus

White spot syndrome virus (WSSV) is an enveloped double-stranded DNA virus that infects a wide range of aquatic crustaceans and thus can have a severe economic impact. Liu and Yang [40] identified ORF ws112 a gene encoding a protein of 461 amino acids as a putative dUTPase. Sequence comparisons demonstrated that the C-terminal end of the protein exhibited low similarity to other known proteins, while the N-terminal end contained the highest sequence similarity to several homotrimeric dUTPases with the highest similarity being with the fowl adenovirus dUTPase. Cloning and expression of the N-terminal region resulted in a 176 amino acid protein that possessed classical dUTPase activity. Structural studies have demonstrated that the WSSV dUTPase like the ASFV dUTPase, while homotrimeric, have active sites composed of two subunits [41]. While it has been suggested that the WSSV dUTPase functions to maintain genomic integrity during virus replication, there have not been any further studies to determine how the C-terminal end of the full-length protein may impact the function of the dUTPase in vivo.

### 3.4. Poxviridae

The *Poxviridae* is a large family of double-stranded DNA-containing viruses classified into two subfamilies; *Chordopoxvirinae* whose members infect vertebrates and those in the subfamily *Entomopoxvirinae*, whose members infect insects. Putative *dUT* genes have been identified in the *Avipoxvirus*, *Cervidpoxvirus*, *Leporipoxvirus*, *Orthopoxvirus*, *Parapovirus*, *Suipoxvirus* (swinepox) and *Yatapox* genera based upon DNA sequences and annotation (Table 1) Molluscum contagiosum, a member of the *Mulluscipox* and a potential human pathogen, has been reported to lack a *dUT* encoding gene [42]. The gene encoding the dUTPases from Orf, a parapoxvirus and vaccinia an orthopoxvirus have been cloned, expressed and reported to encode for dUTPase enzymatic activity (Table 2) [43,44,45]. The crystal structure of the vaccinia virus dUTPase protein revealed that it is a homotrimer [46]. Interestingly, vaccinia virus and myxovirus are being developed for use as delivery vectors and potential oncolytic virotherpeutics. While the myxovirus is replication-incompetent in humans, the use of replication-competent vaccinia virus could express the poxvirus dUTPase in humans, if used as a vaccine. However, there have not been any studies to address the potential effect that this may have in the host.

### 3.5. Retroviridae

*Retroviridae*, whose members are placed in two subfamilies *Orthoretrovirinae* and *Spumaretrovirinae*, are single-stranded RNA viruses with a DNA intermediate in their life cycle. Elder et al. [47] first reported that distinct subsets of non-primate lentivirus virus, but not primate lentiviruses, as well as type D retroviruses contained a gene segment potentially encoding for a dUTPase. Subsequently, several studies using feline immunodeficiency virus (FIV), equine infectious anemia virus (EIAV), Mason–Pfizer monkey virus (MPMV) and simian type D retrovirus demonstrated that these viruses encoded for a protein exhibiting dUTPase enzymatic activity [48,49,50,51] (Table 2). Putative dUTPase genes have also been identified in endogenous retrovirus (ERV) genomes. Endogenous retroviruses are not formally included in retrovirus classification system, and are classified broadly on the basis of relatedness to exogenous genera. Class II ERVs are most similar to the betaretroviruses and alpharetroviruses and putative dUTPase-encoding genes have been reported in the class II betaretrovirus ERVs (Table 1).

Jern et al. [21] reported that acquisition of dUTPase occurred three times in retroviral evolution, which is reflected in the location of the gene encoding the dUTPase with the retrovirus genome. In non-primate lentiviral genomes the dUTPase gene is located within the polymerase (*Pol*) gene, while the betaretroviral dUTPase gene is located in the N-terminal of the protease-encoding region. A third dUTPase acquisition event occurred in the endogenous retrovirus murine ERV-L (MuERV-L) [52], where the dUTPase-encoding region is located C-terminal of the integrase gene. 

The biochemical and structural characteristics of several exogenous retroviruses (Mouse Mammary Tumor Virus (MMTV), Mason–Pfizer Monkey Virus (MPMV), Caprine Arthritis Encephalitis Virus (CAEV), Equine Infectious Anemia Virus (EIAV) and Feline Immunodeficiency Virus (FIV)), as well as an endogenous retrovirus (human endogenous retrovirus-K), have been reported [48,49,50,51,53,54,55,56,57]. These dUTPase proteins contain the ordered series of conserved motifs (N-terminal to C-terminal; I, II, III, IV, and V), characteristic of dUTPases, and they also exist as homotrimers. 

An unusual feature of some of the retroviral dUTPase proteins is that they are located in the virion, either as an independent protein (CAEV, EIAV, FIV and Maedi–Visna virus) [48,49,55] or as a fusion protein with the nucleocapsid protein that is localized within the virion (MMTV and MPMV) [53,54]. 

Most studies of the dUTPase protein encoded by exogenous retroviruses have focused on its role in promoting the replication of the virus in various cell types. These studies have demonstrated that the dUTPase is required for replication of the retroviruses in various differentiated cells, which these viruses infect. This is thought to be due to the enzymatic activity of the viral dUTPase, since cellular dUTPases are in low abundance in differentiated cells. 

### 3.6. Herpesvirales

This is an order of large linear double-stranded DNA viruses characterized by an icosahedral capsid enclosed in an envelope. The order is made up of three families—*Alloherpesviridae*, whose members infect fish, *Malacoherpesviridae*, whose members infect mollusks and *Herpesviridae*, whose members infect birds, mammals and reptiles. The *Herpesviridae* is subdivided into three subfamilies based upon biological and genomic features; the Alpha-, Beta-, and Gammaherpesvirinae [58]. Members of the *Herpesviridae* exhibit at least two biocycles, a lytic (productive) cycle in which new viruses are formed and a latent cycle in which few if any viral genes are expressed. The ability to establish latency in a host ensures a lifelong infection. While latency is thought to occur in members of the *Alloherpesviridae* and *Malacoherpesviridae*, this has not been confirmed. There are approximately 130 species distributed between the three subfamilies.

Genes encoding for putative dUTPases have been identified in members of the *Alloherpesviridae* [59,60,61] and *Malacoherpesviridae* [62,63,64] (Table 1). Annotation of these genes indicates that they encode for proteins that exhibit the common motif order (I, II, III, IV, and V) suggesting a homotrimeric structure. This arrangement is consistent with phylogenetic studies suggesting that these viruses may have acquired the gene independently or early in the evolution of herpesviruses, their origin being estimated at 180–200 million years ago, before the duplication observed in Alpha- and Gammaherpesvirus lineages. Of note, three potential *dUT* genes have been identified in ostreid herpesvirus 1, a virus that infects and causes a high mortality rate in Pacific oysters [65]. It was predicted that ORF75 encoded an enzymatically functional dUTPase while ORF 27 and 34 encoded for inactive proteins. However, the potential functions of the proteins encoded by ORFs 27 and 34 remain unknown.

Of those dUTPase genes examined in the *Herpesviridae*, they are only expressed during lytic (productive) replication. Early studies into lytic replication of these viruses demonstrated that the expression of lytic genes occurred in a sequential cascade manner, with immediate early genes encoding for transcriptional factors that drive replication, early genes encoding for proteins involved in DNA replication and late genes which encoded for structural proteins. While more recent studies have demonstrated that the expression of some genes during lytic replication do not behave in this ordered fashion with the exception of the human cytomegalovirus (HCMV) UL72 gene, dUTPase-encoding genes are expressed as early genes. 

While several putative *dUT* genes have been identified in species in the Alpha- [66,67,68,69,70,71] and gammaherpesvirus [72,73,74] lineages, most studies have focused on members that infect humans primarily because these viruses cause more morbidity in humans than any other group of viruses except those that cause the common cold. These viruses include herpes simplex virus 1 and 2 (HSV-1 and -2), the causative agents of herpes labialis and genital herpes, respectively [75,76,77], and varicella-zoster virus (VZV), the etiological agent of chickenpox and shingles [78] of the *Alphaherpesvirinae*, the *Betaherpesvirinae* HCMV, HHV-6 and 7, which are associated with various diseases in transplant patients and the *Gammaherpesvirinae* Epstein–Barr virus, which is the etiological agent of infectious mononucleosis and is associated with several human malignancies including Burkitt’s lymphoma, nasopharyngeal carcinoma, and Hodgkin’s lymphoma [79,80] and human herpesvirus 8 (Kaposi’s sarcoma-associated herpesvirus; HHV-8), which is associated with Kaposi’s sarcoma, primary effusion lymphoma and multicentric Castleman’s disease [81]. However, analyses have demonstrated that the conserved motifs arrangement in these proteins is III, I II, IV and V, indicating that they are monomeric in structure. However, a crystal structure has only been determined for EBV [14] [Table 1 and Table 2].

Members of the *Betaherpesvirinae* subfamily which have been sequenced contain a genomic locus corresponding to the loci of *dUT* genes in members of the *Alphaherpesvirinae and Gammaherpesvirinae* subfamilies [82]. While these genes have been referred to as putative *dUT* genes, annotation demonstrated that the proteins from these genes lacked the classical dUTPase motifs I–V. Furthermore, cloning of the UL72 gene from HCMV [24] and the U45 gene from human herpesvirus 6 A (HHV-6) [25] demonstrated that the recombinant proteins lacked detectable dUTPase enzymatic activity. Interestingly, McGeehan et al. [20] reported that the location of the original motif III is occupied by a conserved motif VI in the *Alphaherpesvirinae and Gammaherpesvirinae* and is also present in the orthologous proteins in the *Betaherpesvirinae*. The function of motif VI is unknown. 

An interesting feature of some members of the *Alphaherpesvirinae* (HSV-1, HSV-2, PRV, and VZV) and *Betaherpesvirinae* (HCMV) is the that the dUTPase protein is a tegument protein and thus is located in the virion, which distinguishes these viruses from those of the *Gammaherpesvirinae* members [83,84,85,86,87,88]. 

## 4. Additional Functions of Viral dUTPase Proteins

### 4.1. Aviadenovirus

Fowl aviadenoviruses (FAdVs) are distributed worldwide in poultry farms. Some FAdVs are the causative agents of inclusion body hepatitis and hydropericardium syndrome, which cause significant economic losses to the poultry industry. Using genomic sequence analyses, Deng et al. [89] reported that of the fowl adenovirus (FAdV) serotypes examined, all contained a gene (ORF 1) that encodes for a dUTPase based upon the demonstration of the five conserved motifs characteristic of this protein. Furthermore, cloning of ORF1 from FAdV serotype 9 demonstrated that the gene encoded for polypeptide of 163 amino acids that possessed functional dUTPase enzymatic activity. Somewhat interesting, these investigators also reported that the FAdV dUTPase in addition to classical enzymatic activity, the protein also upregulated the expression of interferon α and β in vitro. In a follow-up study using a *dUT* knock-out virus in vivo, these investigators reported that the dUTPase protein was modulating the innate immune response against the virus [90]. There have not been any additional studies to address the role of this protein in FAdV pathogenesis.

### 4.2. Exogenous Retroviruses

CAEV is a small ruminant lentivirus in the family *Retroviridae*, which is the etiological agent of caprine arthritis-encephalitis, a chronic progressive disease primarily in goats. Recent studies have reported that the dUTPase protein of CAEV inhibits IFN β production, allowing for the increased replication of Sendi virus and vesicular stomatis virus in vitro [91,92]. Shi et al. demonstrate that the CAEV dUTPase inhibited IFN β production by altering signal transduction pathways upstream of interferon regulatory factor 3 (IRF3) [92].

### 4.3. Endogenous Retroviruses

Endogenous retroviruses (ERVs) are sequences within the genome of the host that represent an ancient infection of a germline cell with an exogenous retrovirus, which is passed vertically to progeny and inherited as a Mendelian gene. ERVs are currently grouped into three classes based upon their phylogenetic relatedness to exogenous viruses; class I Gammaretrovirus- and Epsilonretrovirus-like (HERV-E, -F, -FRD -H, -I, -P, -R, -T, and -W), class II Betaretrovirus-like (HERV-K) and class III comprises elements most similar to Spumaretrovirus-like (HERV-L).

Human endogenous retroviruses (HERVs) constitute 7–8% of the human genome and several different families exist. Most HERVs are defective due to recombination events and the accumulation of nonsense mutations, ensuring that no protein or viable virus particles are produced. Conversely, members of the HERV-K family, which make up less than 1% of the HERV sequences found in the human genome, are transcriptionally active. The conservation of HERV in the human genome could provide promoters, enhancers, repressors, poly-A signals and alternative splicing sites for human gene transcripts. The pathogenicity of exogenous retroviruses resulted in numerous studies to discover a correlation between HERVs and different human diseases such as cancer, multiple sclerosis and autoimmune diseases. However, until recently, there were no studies that demonstrated a causal relationship between HERVs and human disease.

Based upon DNA sequence comparisons, Harris et al. [93] demonstrated that HERV-K encoded for a dUTPase protein that lacked enzymatic activity. Using site-directed mutagenesis approaches of a HERV-K consensus dUTPase DNA sequence, they restored enzymatic activity to the protein which exhibit properties similar to those of dUTPases. They suggested that this dUTPase corresponded to the ancestral, wild-type DNA sequence. In a subsequent study [57], they reported that the HERV-K dUTPase exhibited a homotrimeric structure that was similar to the human dUTPase.

#### 4.3.1. Psoriasis

Psoriasis is a chronic immune-mediated inflammatory skin disorder, which carries an increased risk for extracutaneous comorbidities such as psoriatic arthritis and cardiovascular disease. Using haplotype sharing analyses, Foerster et al. [94] reported that the presence of a gene encoding a HERV-K dUTPase, mapping within a 60 kb region telomeric to HLA-C at chromosome 6p21 (1) referred to a PSOR1, confers increased susceptibility to psoriasis. Using a recombinant HERV-K dUTPase protein based upon the consensus DNA sequence, it was demonstrated that the protein, which lacks enzymatic activity, induced the activation of NF-κB through Toll-like receptor 2 (TLR2). Proteome array studies revealed that treatment of human primary dendritic/Langerhans-like cells and keratinocytes with HERV-K dUTPase protein triggered the secretion of Th1 and Th17 cytokines involved in the formation of psoriatic plaques, including IL-12p40, IL-23, IL-17, tumor necrosis factor-α, IL-8 and CCL20 supporting the premise of HERV-K dUTPase as a potential contributor to psoriasis pathophysiology [95]. In a subsequent study using a candidate gene approach sequencing 708 psoriasis cases and 349 controls, it was reported that the HERV-K dUTPase variants are strongly associated with psoriasis [96]. Sequencing of the HERV-K PSORS dUTPase region demonstrated that the gene encoded a fusion protein whose N-terminal 93 amino acids corresponded to a nucleocapsid (NC) protein while the remaining 116 amino acids contained the motifs of classical dUTPases. This is similar to the NC-dUTPase protein in MPMV. 

Gupta et al. [97] reported that the increased humoral response to the HERV-K dUTPase in psoriasis was probably due to cross-reactivity with the *Candida albicans* encoded protein Ca019.10692. Notably, they employed a peptide (KCYNCGQIGHLKKNC, peptide-137), 15 amino acids in length, derived from the nucleocapsid domain of gag that has no similarity to the HERV-K dUTPase and thus their experimental data do not support this conclusion. 

#### 4.3.2. Pulmonary Arterial Hypertension (PAH)

PAH is a progressive disorder characterized by endothelial cell (EC) dysfunction, loss of distal pulmonary arteries (PAs), and destructive changes in more proximal PAs in association with expansion of cells that progressively occlude the vessel lumen. These features contribute to an elevation in right ventricular systolic pressure, which, despite vasodilator therapy, can lead to right heart failure and the need for a lung transplant. Inflammatory and autoimmune processes are inextricably linked to vascular remodeling in PAH.

Recently, the elevation of SAM domain- and HD domain-containing protein 1 (SAMHD1), an innate immune factor, in immune complexes of patients with hereditary and idiopathic PAH has been demonstrated. This elevation is due in part to the increased expression of HERV-K dUTPase in circulating monocytes and pulmonary arterial (PA) adventitial macrophages of these patients [98]. Furthermore, using a rat model, these investigators reported that the HERV-K dUTPase activated B cells, elevated pro-inflammatory cytokines IL-6, interleukin 1β (IL-1β), and tumor necrosis factor-α (TNF-α) in monocytes and pulmonary arterial endothelial cells (PAECs), increased pulmonary artery vulnerability to apoptosis, and induced PAH, suggesting that the HERV-K dUTPase contributed to sustained inflammation, immune dysregulation, and progressive obliterative vascular remodeling that occurs in PAH. In a follow-up study, Otsuki et al. reported that monocytes overexpressing HERV-K dUTPase released HERV-K dUTPase in extracellular vesicles (EVs), causing pulmonary hypertension in mice in association with endothelial mesenchymal transition (EndMT) related to the induction of the transcription factors SNAIL/SLUG, and pro-inflammatory molecule IL-6 as well as vascular cell adhesion molecule 1 (VCAM1) [99]. In PAECs, HERV-K dUTPase required Toll-like receptor 4 (TLR4)-myeloid differentiation primary response (MYD)-88 to increase IL-6 and SNAIL/SLUG, and HERV-K dUTPase interaction with melanoma cell adhesion molecule (MCAM) was necessary to upregulate VCAM1. Thus, it was concluded in PAH, monocytes or macrophages released HERV-K dUTPase in EVs, and HERV-K dUTPase engaged dual receptors and signaling pathways to subvert PAEC transcriptional machinery to induce EndMT and associated pro-inflammatory molecules. These results further confirm that the HERV-K dUTPase protein has novel immunological functions, in the absence of enzymatic activity, which can contribute to disease processes. 

### 4.4. Alphaherpesviriniae

Pseudorabies virus (PRV; Suid alphaherpesvirus 1) is the causative agent of Aujeszky’s disease in pigs. In younger suckling pigs, the mortality rate is close to 100% [100]. The UL50 gene encodes for a dUTPase that has functional enzymatic activity [101] (Table 2). Additional studies by these investigators reported that PRV mutants defective in UL50 conferred protection to pigs in a challenge assay using wild-type PRV, suggesting that the UL50 mutants could be used as a vaccine [102]. A study to identify proteins that antagonize type 1 interferon (IFN) signaling [103] reported that the dUTPase protein encoded by UL50 of PRV and HSV-1 inhibited type I IFN-induced STAT1 phosphorylation, likely by accelerating lysosomal degradation of IFN receptor 1 (IFNAR1). The inhibition of type 1 IFN signaling was independent of its dUTPase enzymatic activity and in the case of PRV dUTPase protein, it required amino acids 225 to 253 in the C-terminal region. 

### 4.5. Gammaherpesvirinae

#### 4.5.1. Murid Gamma Herpesvirus 68

Murine gammaherpesvirus 68; MHV-68 is a natural pathogen of wild murid rodents and is capable of infecting both outbred and inbred mice [104]. Leang et al. [105] reported that ORF54, which encodes for an enzymatically functional dUTPase, induced the degradation of the type 1 interferon receptor protein IFNAR1. The degradation of IFNAR1 was independent of the dUTPase enzymatic function. Furthermore, mutants lacking functional ORF54 were unable to establish latent infection in lymphocytes. 

#### 4.5.2. HHV-8

Madrid et al. [106] reported that the dUTPase protein encoded by ORF54 downregulated the expression of NKp44L, an activating receptor present on natural killer (NK) cells. The downregulation of NKp44L was independent of the dUTPase enzymatic activity. The dUTPase protein homologues from EBV (BLLF3) and HSV-1 (UL50) did not downregulate NKp44L expression. Furthermore, they reported that the dUTPase protein also downregulated the expression several cytokine receptors, including IFNAR1, gp130, interleukin-23 receptor (IL-23R), and interferon gamma receptor 1 (IFNGR1), suggesting that the mechanism involves perturbation of membrane trafficking. ORF54 appears to function by altering the subcellular localization of NKp44L molecules, relocating them from the cell surface to intracellular compartments. Farhat [107] reported that the HHV-8 dUTPase protein facilitated the selective degradation of IFNAR1 mediated by the endosomal sorting complexes required for transport (ESCRT) machinery, which distinguishes the HHV-8 dUTPase protein ORF54 from other known viral mechanisms that induce IFNAR1 degradation.

### 4.6. Alphaherpesviriniae, Betaherpesvirinae and Gammaherpesvirinae

It has been demonstrated fairly conclusively that the human herpesviruses dUTPase proteins possess non-canonical immune and neurological functions that may contribute to the pathologies observed in diseases associated with these viruses. The human herpesviruses dUTPase proteins act as Pathogen-Associated Molecular Pattern (PAMP) molecules and, through their ligation with Toll-like receptor (TLR)2 homodimers in the case of the EBV dUTPase or TLR2/1 heterodimers in the case of the HSV-2, HHV-6, HHV-8 and VZV dUTPases, differentially activate NF-κB, with subsequent modulation of downstream genes involved in chronic inflammation, innate and adaptive immune modulation and neurotransmitter function [25,108,109,110,111,112,113,114]. In addition to increasing the expression/secretion of the proinflammatory cytokines IL-1β, IL-6, IL-8, IL-12, TNF-α, and IL-10 and IFN-γ from human dendritic cells, macrophages and PBMCs, studies with the EBV dUTPase protein have demonstrated that it is released in extracellular vesicles [111], modulates the differentiation and proliferation of follicular helper T cells, a cell type critical for germinal center development [112], and alters the expression of in vitro and in vivo of targets with central roles in blood–brain barrier integrity/function, fatigue, pain, synapse structure and function, as well as tryptophan, dopamine, and serotonin metabolism [114].

## 5. dUTPase Sequences Conserved in Other Viral Genes

### 5.1. Adenoviridinae: Mastadenovirus

The genus of adenovirus responsible for human infections is *Mastadenovirus.* This group is estimated to be responsible for 2–5% of respiratory infections as well as gastrointestinal and eye infections. The adenovirus E4 ORF1 genomic region encodes for a 125 amino acid protein that has been implicated as an oncogene (adenovirus serotype 9) [115] and as being responsible for inducing adipogenesis (adenovirus 36) [116]. 

E4 ORF1 appears to be derived from a dUTPase gene, but its descendants in human adenoviruses have not retained the active site residues and presumably carry out other functions [116]. By performing pairwise sequence alignments between full-length E4 ORF1 and dUTPase sequences (*Homo sapiens*, *Candida albicans*, *Saccharomyces cerevisiae*, *Lycopersicon esculentum*, avian adenovirus CELO, and poxviruses (vaccinia, variola, and Orf)), it was determined that the sequence similarities between E4 ORF1 and dUTPase proteins extended over the entire lengths of the E4 ORF1 proteins [117]. Evidence supports the idea that adenovirus E4-ORF1 genes evolved from an ancestral dUTPase gene and functionally diverged over time [116,117] (Table 3).

### 5.2. Retoviridae: Human Immunodeficiency Virus

Human immunodeficiency virus (HIV) envelop glycoprotein gp120 is required for the interaction of the virion with the CD4 receptor on cells permissive for viral replication. gp120 also binds to several other receptors expressed by non-permissive cells including TLR2. Activation of these receptors is mediated by viral particles or soluble gp120, which is secreted or released following lysis of infected cells [131]. Binding of gp120 to TLR2 results in the induction of several cytokines/chemokines that influence the host’s immune response to the virus [118,119,131,132,133]. Interestingly, Abergel et al. [134] reported the presence of weak but significant sequence similarity between gp120 and the human dUTPase. A previous study had suggested structural similarity between the variable loop 3 (V3) region and the *Escherichia coli* dUTPase [135]. The human and *E. coli* dUTPases are prototypical dUTPases exhibiting the classical ordered series of conserved motifs (N-terminal to C-terminal; I, II, III, IV, and V) and trimeric structure, gp120 also exist as a trimeric structure. Furthermore, the V3 loop of gp120 is critical for binding of CCR5 and CXCR4 the chemokine receptors necessary for HIV infection [136,137]. Thus, it was suggested that an ancestral dUTPase gene evolved into the present primate lentivirus CD4 and cytokine receptor interacting region of gp120. 

### 5.3. Betaherpesvirinae

Computational analysis of the coding sequences of several genes of HCMV revealed dUTPase-related motifs specifically motif VI [23]. The UL31 gene encodes for a protein (pUL31) of 595 amino acids containing a C-terminal dUTPase-like structure. The UL31 gene is expressed as a late gene and pUL31 is localized to nucleolin-containing nuclear domains where it is required to reduce pre-ribosomal RNA (rRNA). This modulation of pre-rRNA was dependent on the dUTPase-like motif in pUL31 [138]. In addition to modulating pre-rRNA, pUL31 has been reported to interact with cyclic GMP-AMP synthase (cGAS), a cytosolic DNA sensor that recognizes virus DNA, preventing downstream signaling events leading to the induction of type 1 interferon (IFNα/β) and antiviral effector genes [120]. The UL82 gene encodes for a 559 amino acid protein referred to as pp71, which is a major tegument protein. Similar to pUL31, pp71 modulates the cGAS signaling cascade by inhibiting the translocation of stimulator of interferon genes (STING) and the recruitment of TBK1 and IRF3 to the STING complex, preventing downstream signaling events leading to the induction of type 1 IFN and antiviral effector genes [121]. The UL83 gene encodes for a pp65 a 561 amino acid tegument phosphoprotein. The phosphorylated pp65 inhibits type 1 IFN signaling by preventing the trafficking of interferon regulatory factor 3 (IRF3) to the nucleus [122]. The UL84 gene encodes for a 586 amino acid protein referred to as UL84 protein, which has an essential role in the initiation of viral DNA replication by interacting with the origin of lytic replication, oriLyt, and then recruiting other viral/cellular factors required for replication [123].

### 5.4. Gammaherpesvirinae

#### 5.4.1. EBV: LF1 and LF2 Genes

The LF1 (ORF10) and LF2 (ORF11) genes are hypothesized to have arisen from a gene duplication event of the BLLF3 gene of EBV [23]. BLLF3 encodes for a dUTPase. The region of homology between the LF2 protein and BLLF3 comprises motif VI, which is unique to the dUTPases encoded by members of the *Herpesviridae.* LF1 encodes for a 469 amino acid protein (G10) of unknown function. LF2 encodes for a 429 amino acid protein (protein LF2) that has been reported to interact with IRF7, suppressing IFN-α production [124] and to interact with BRLF1 (Rta) regulating EBV replication [125].

#### 5.4.2. HHV-8: ORF10 and ORF 11

The ORF10 and ORF 11 of HHV-8 genes are hypothesized to have arisen from a dUTPase [23]. ORF10 encodes for a 418 amino acid protein that has been reported to block interferon signaling by forming complexes that contain IFNAR subunits and the Janus kinases Jak1 and Tyk2 [126] and to inhibit mRNA export of cellular transcripts by interacting with the nuclear RNA export factor Rae1 [127].

Computational analysis of the ORF11 coding sequence revealed a dUTPase-related domain, specifically motifs I, II, IV and VI [23], suggesting that the KSHV ORF11 gene product may function as a dUTPase. However, cloning of the gene with subsequent protein production demonstrated that the ORF11 lacked dUTPase activity [128]. ORF11 encodes for a 407 amino acid protein that exhibits some homology with the EBV LF2 protein. It is also reported to be located in the tegument region of the virion [129,130].

## 6. Concluding Remarks

dUTPases are ubiquitous enzymes found in most free-living organisms as well as many viruses. The primary focus of studies on the family of enzymes has been on their role in maintaining genome integrity. However, there is increasing evidence suggesting that dUTPases encoded by various viruses may have unexpected and novel functions in regulating the host’s innate immune response. Infections by these viruses in various mammalian populations can have a severe economic impact and cause excessive morbidity and mortality. While additional studies are required, a thorough understanding of the novel properties of the viral dUTPases may lead to the development of vaccines and antiviral therapies for use against infections caused by dUTPase-expressing viruses. 

## Figures and Tables

**Figure 1 biomolecules-12-00227-f001:**
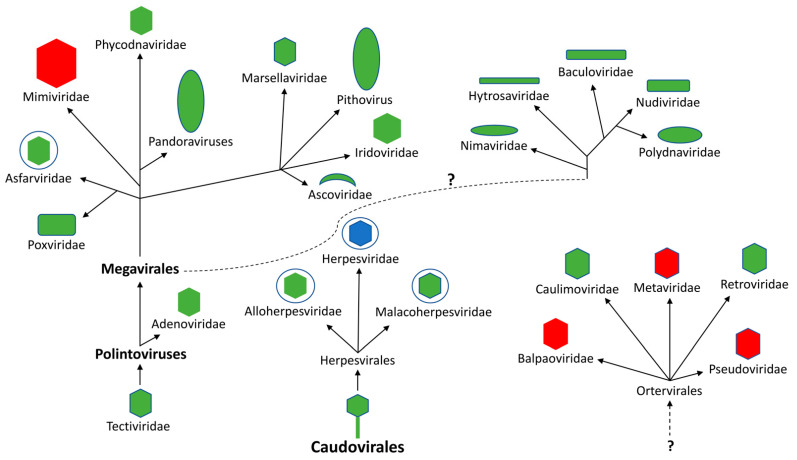
Evolutionary distribution of putative *dUT* genes in double-stranded DNA viruses and single-stranded RNA viruses that replicate through a DNA intermediate (*Ortervirales*). The DNA viruses are proposed to have evolved from two distinct groups of bacteriophages (*Tectiviridae* and *Caudovirales*) [15]. The dotted line with a question mark indicates an unknown evolutionary relationship. Red indicates that no members have been reported to possess putative *dUT* gene, green indicates that at least one member has been reported to contain a *dUT* gene encoding a homotrimeric protein, while blue indicates that a least one member contains a *dUT* gene encoding a monomeric protein.

**Figure 2 biomolecules-12-00227-f002:**
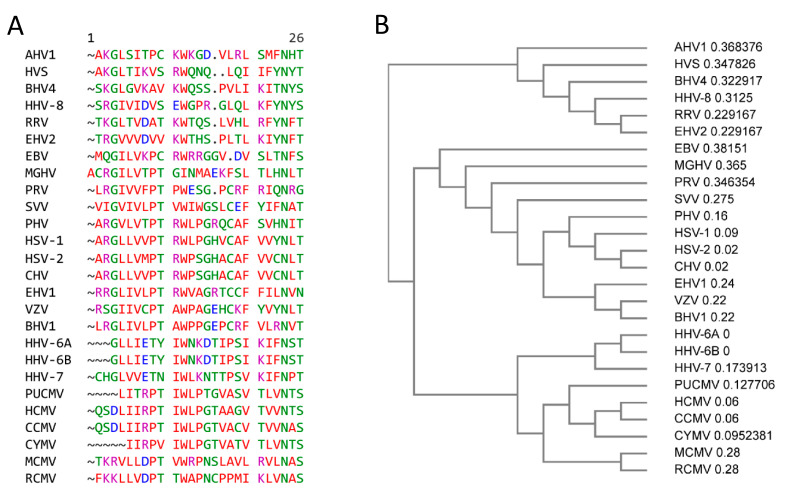
Motif VI of members of the *Herpesvirales*. (**A**) CLUSTAL OMEGA multisequence alignment of herpesvirus dUTPase Motif IV (Genebank accession code and amino acid (aa) length of motif VI) HSV-1; herpes simplex 1 (NC_001806.2, aa 265–289); HSV-2: herpes simplex 2 (NC_001798.2, aa 263–287); CHV: chimpanzee alphaherpesvirus (NC_023677.1, aa 263–297); PHV Papiine alphaherpesvirus 2 (NC_007653.1, aa 259–283); VZV: Varicella-Zoster virus (NC_001348.1, aa 205–225); SVV: simian varicella virus (NC_002686.2, aa 285–309); EHV-1: equine herpesvirus 1 (AY665713.1, aa 223–247); BHV1: bovine herpesvirus 1 (KU198480.1, aa 223–247); PRV: pseudorabies virus (BK001744.1, aa 177–200); HCMV: human cytomegalovirus (NC_006273.2, aa 266–291); HHV-6A: human herpes virus-6A (KP257584.1, aa 300–322); HHV-6B: human herpesvirus 6B (NC-000898.1, aa 300–322); CCMV: chimpanzee cytomegalovirus (NC_003521.1, aa 251–278); CYMV: Cynomolgus cytomegalovirus (KP796148.1, aa 259–279); RCMV: rat cytomegalovirus (NC_002512.2, aa 270–285); EBV: Epstein–Barr virus (AJ507799.2, aa 176–198); HHV-8: human herpesvirus 8 (NC_009333.1, aa 206–229); RRV: rhesus rhadinovirus (NC_003401.1, aa 181–204); HVS: herpesvirus saimiri (NC_001350.1, aa 178–200); MGHV: Murine gammaherpesvirus 68 (AF105037.1, 185–210); BVHV: bovine herpesvirus 4 (NC_002665.1, aa 176–200); EHV2: equine herpesvirus 2 (NC_001650.2, aa 181–204); AHV1: alcelphine herpesvirus 1 (AF005370, aa 185–208). (**B**) Phylogram guide tree of herpesviruses motif VI.

**Table 1 biomolecules-12-00227-t001:** Putative dUTPases based on DNA sequencing and annotation.

Family	Virus	ORF	Amino Acids	Predicted Structural Family
Adenoviridae				
	Aviadenovirus	104L	149	Trimer
	Tupaia(tree-shrew) adenovirus			
	NC_044936.1			
Ascoviridae				
	Diachasminimorpha longicaudata entomopoxvirus (DIEPV)	DELV007/187	151	Trimer
	KR095315.1			
Iridoviridae Alphairidovirinae				

	Frog virus 3	63R	163	Trimer
	NC_005946.1			
	Tiger frog virus	68R	163	Trimer
	MT512504.1			
	Ambystoma tigrinum virus	42L	144	Trimer
	MK580533.2			
	Grouper Iridovirus	28L	159	Trimer
	AY666015.1			
	Singapore grouper Iridovirus	49L	155	Trimer
	NC_006549.1			
Iridoviridae Betairidovirinae				

	Iridovirus type 6	438L	296	Trimer
	NC_003038.1			
	Iridovirus type 9	45R	188	Trimer
	MK500311.1			
	Chameleon Iridovirus	Upstream of 438L	349	N-terminal domain splicing and C-terminal dUTPase domain
	MN081869.1			
	Lizard-cricket iridovirus variant IV	438L	296	Trimer
	MN081869.1			
	Lizard-cricket iridovirus variant IV	438L	478	Second dUT gene adjacent to 438L
	MN081869.1			
Nudiviridae				
	Dikerogammarus haemobaphes nudivirus	009	231	Trimer
	MT488302.1			
	Helicoverpa zea nudivirus 2	H22V69	350	Trimer
	NC_JN4188.1			
Poxviridae				
	Variola	F2L	147	Trimer
	X69198.1			
	Monkeypox	C8L	151	Trimer
	KJ642619.1			
	Camelpox	DUT	147	Trimer
	AF438165.1			
	Bovine papular stomatitis virus	007	163	Trimer
	NC_005337.1			
	Tanapox virus	17L	143	Trimer
	AF153912.1			
	Yaba monkey tumor virus	17L	143	Trimer
	NC_005179.1			
	Lumpy skin disease virus	018	141	Trimer
	NC_003027.1			
	Myxoma virus	M012L	148	Trimer
	KP723391.1			
	Shope fibroma virus	S012L	143	Trimer
	NC_001266.1			
	Swinepox	SPV013	141	Trimer
	MZ773480.1			
Baculoviridae Alphabaculovirus				

	Lymantria dispar multiple nucleopolyhedrovirus	116	149	Trimer
	MF311096.1			
	Spodoptera eridania nucleopolyhedrovirus isolate 251	61	144	Trimer
	NC_055502.1			
	Spodoptera exigua multiple nucleopolyhedrovirus	55	143	Trimer
	NC_002169.1			
	Spodoptera frugiperda nucleopolyhedrovirus	54	144	Trimer
	MZ292981.1			
Baculoviridae Betabaculovirus				

	Spodoptera litura nucleopolyhedrovirus II	10	164	Trimer
	AF325755.1			
	Helicoverpa armigera multiple nucleopolyhedrovirus	65	142	Trimer
	NC_010240.1			
	Pseudaletia (Mythimna) sp. granulovirus #8	98	145	Trimer
	NC_033780.1			
Giant Viruses				
	Marseillevirus	015	71 (truncated)	Trimer
	NC_013756.1			
	Medusavirus	348	164	Trimer
	MW018138			
	Pandoravirus salinus	1148	212	Trimer
	NC_022098.1			
	Ostreococcus tauri virus	OtV-1_199	142	Trimer
	NC_013288.1			
Retroviridae				
	Bovine immunodeficiency virus	Polymerase reading frame	74	Truncated, no enzymatic activity structure unknown
	M32690.1			
Alloherpesviridae				
	Anguillid herpesvirus 1	5	168	Trimer
	NC_013668.3			
	Cyprinid herpesvirus 2	123	156	Trimer
	MN201961.1			
	Salmonid herpesvirus 1	49	177	Trimer
	AF023673.1			
Malacoherpesviridae				
	Ostreid herpesvirus 1	27	266 (inactive)	Trimer
	MW412420.1	34	124 (inactive)	
		75	236 (active)	
Alphaherpesvirinae				
	Gallid alphaherpesvirus 2	UL50	395	Monomer
	MC518371.1			
	Testudinid alphaherpesvirus 3	74	448	Monomer
	NC_027916.2			
	Equine herpesvirus 1	9	327	Monomer
	AY665713.1			
	Bovine herpesvirus 1	UL50	325	Monomer
	KU198480.1			
Gammaherpesvirinae				
	Bovine gammaherpesvirus 6	54	290	Monomer
	NC_024303.1			
	Equid gammaherpesvirus 2	54	289	Monomer
	NC_001650.2			
	Saimiriine gammaherpesvirus 2	54	301	Monomer
	NC_001350.1			
	Alcelaphine herpesvirus 1	54	298	Monomer
	KX(05134.1)			

**Table 2 biomolecules-12-00227-t002:** Validated dUTPases.

Family	Virus	ORF	Amino Acids	Predicted Structural Family
Adenoviridae				
	Fowl adenovirus	1	163	Trimer
	KT862812.1			
Asfarviridae				
	African swine fever virus	E165R	165	Trimer but only has two catalytic sites
	NC_044945.1			
Iridoviridae				
	Rana grylio virus	DUT	164	Trimer
	JQ654586			
Nimaviridae				
	White spot syndrome virus	Ws112	461	Trimer: Fusion protein, dUTPase in N-terminal, C-terminal function unknown
	KY827813.1			
Poxviridae				
	Vaccinia	FL2	147	Trimer
	NC_006998.1			
	Orf	E3L	159	Trimer
	MT332357.1			
Retroviridae				
	Mouse mammary tumor virus	Protease reading frame	134	Trimer: Fusion with nucleocapsid protein, location virion
	NC_001503.1			
	Mason–Pfizer monkey virus	Protease reading frame	154	Trimer: Fusion with nucleocapsid protein, location virion Crystalized
	NC_001550.1			
	Caprine arthritis encephalitis virus	Polymerase reading frame	168	Trimeric structure: virion
	GU120138.1			
	Equine infectious anemia virus	Polymerase reading frame	136	Trimeric structure virion: Crystalized
	M16575.1			
	Feline immunodeficiency virus	Polymerase reading frame	133	Trimeric structure virion: Crystalized
	M25381.1			
	Maedi–Visna virus	Polymerase reading frame	168	Trimeric structure: virion
	MW248464.1			
Retroviridae: Human Endogenous Retroviruses				
	Human endogenous retrovirus-K	Multiple chromosomal sites	171	Trimeric structure: Crystalized
	JQ966584.1			
Alphaherpesvirinae				
	Gallid herpesvirus 1	UL50	486	Monomer
	JN542536			
	Herpes simplex virus-1	UL50	372	Monomer virion
	NC_001806.2			
	Herpes simplex virus-2	UL50	369	Monomer virion
	NC_001798.2			
	Pseudorabies virus	UL50	268	Monomer virion
	BK001744.1			
	Varicella-Zoster virus	8	396	Monomer virion
	NC_001348.1			
Betaherpesvirinae *				
	Human cytomegalovirus	UL72	388	Unknown structure
	NC_006273.2			
	Human herpesvirus 6A	U45	376	Unknown structure
	KP257584.1			
	Human herpesvirus 6B	U45	376	Unknown structure
	NC-000898.1			
	Human herpesvirus 7	U45	379	Unknown structure
	NC_001716.2			
Gammaherpesvirinae				
	Epstein–Barr virus	BLLF3	278	Monomer: Crystalized
	AJ507799.2			
	Murine gammaherpesvirus 68	ORF54	299	Monomer
	U97553.2			
	Human herpesvirus 8	ORF54	317	Monomer
	NC_009333.1			

* None of the *Betaherpesvirinae* genes identified encode for a functional dUTPase. Identified based upon DNA sequence comparisons to the *Aphaherpesvirinae* and *Gammaherpesvirinae* and gene location.

**Table 3 biomolecules-12-00227-t003:** dUTPase sequences conserved in other gene families.

Family	Virus	ORF	Function
Adenoviridae			
	Adenovirus 9	E4ORF1	Oncogene [116]
	AF083975.2		
	Adenovirus 36	E4ORF1	Diabetes [117]
	GQ384080.1		
Retroviridae			
	Human immunodeficiency virus	gp120	Ligand for CD4; induces pro-inflammatory cascade, TLR2 triggering [118,119]
	JX245015.1		
Betaherpesvirinae			
	Human cytomegalovirus	UL31	Nucleoli reorganization, inhibits Type 1 IFN production [120,121]
	NC_006273.2		
	Human cytomegalovirus	UL82 (tegument; pp71)	Tegument protein, inhibits cGAS signaling [122]
	NC_006273.2		
	Human cytomegalovirus	UL83 (tegument pp65)	Tegument protein; inhibits Type 1 IFN production [123]
	NC_006273.2		
	Human cytomegalovirus	UL84 (tegument?)	Essential for viral DNA synthesis [123]
	NC_006273.2		
Gammaherpesvirinae			
	Epstein–Barr virus	LF2	Regulates lytic replication and suppresses IFN-α production [124,125,126,127]
	AJ507799.2		
	Human herpesvirus 8	ORF10	Inhibits cell mRNA export and inhibits IFN signaling [128]
	NC_009333.1		
	Human herpesvirus 8	ORF11	Tegument protein, function unknown [129,130]
	NC_009333.1

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
