# Peer review of "Viral dUTPases: Modulators of Innate Immunity"

_biomolecules, 2022, doi:10.3390/biom12020227_

Round 1

Reviewer 1 Report

Please find my review enclosed as a pdf.

Author Response

Reviewer 1:

  1. In the introduction it is important to note that although this enzyme is essential for many living species, still some survive without it (cf. eg. PMID 27933035). At the suggestion of Reviewer 2 the first paragraph has been rewritten in the revised manuscript and the original reference 1 was replaced with the reference suggested by reviewer 1.

  1. I think it would be important to note in the text, how this data has been collected. What is the basis of the list of putative and validated dUTPases, is it a pBLAST search, or Pubmed search or other? A statement “Data used in this manuscript was collected primarily by Pubmed and pBLAST searches.” was included in the revised manuscript.

  1. Lines 97-98 are about Motif 6, I would suggest to include that this motif is located between motif 2 and 4 within the monomeric dUTPases. Of note the appearance of this segment was not perhaps by chance, as to form the dUTPase active site from one polypeptide chain, long additional segments are needed. The location of motif V1 has been included in the revised manuscript.

  1. Lines 129-130.

The predicted amino acid length of these proteins range from 114 to 395 amino acids, suggesting that other non- identified conserved motifs/domains may exist and might be involved in unknown functions of these proteins. Please note that the Genbank entry NC_011615 arguing for a 114 aa-long HearMPNV dUTPase orf65 in PMID: 22913743, has been replaced by NC_004117 arguing for a 142 long dUTPase. The maximum length in Table 1 is supposed to be 478 for CIV (cf below), so the range should be revised. I suggest the authors to add the length of a normal dUTPase as a reference in the text, eg. FIV trimer dUTPase 133 aa-long, EBV monomer dUTPase 278-long. Compared to these there are extended length

DUTs in Table 1. I think it should also be mentioned in this section, that dUTPase itself without any extra domain could exert functions unrelated to its enzymatic activity (cf. HERV-K dUTPase), to balance the concept of seeking for an intriguing new motif.  The revised manuscript has been modified as suggested by the reviewer.  It now reads: “The smallest dUTPases that have been demonstrated to be enzymatically functional are the homotrimeric FIV dUTPase  (133 amino acids) and the monomeric EBV dUTPase (278 amino acids) (Table 2). The predicted amino acid length of dUTPase proteins determined by annotation and from experimental data range from 142 to 478 amino acids, suggesting that other non-identified conserved motifs/domains may exist and might be involved in unknown functions of these proteins.”

  1. When talking about Iridoviruses the reference of Papp et al. PMID: 31269721 is missing. They found a CIV like 266 long dUTPase in orf 438L, and a putative novel 478-aa long dUTPase adjacent to it (cf https://www.ncbi.nlm.nih.gov/nuccore/MN081869) . In Table 1 the orf of Chameleon Iridovirus is 478, which is rather strange. Maybe some revision on this section is needed. In addition in some cases in Table 1 data on both the orf and the length is missing. If in those cases there is no available data please indicate that somehow. The Papp manuscript was not included because there was no specific discussion of the Iridoviridae since the dUT genes are putative based upon DNA sequencing and annotation (see Table 1). If data concerning the gene and/ or the length was not available the virus was omitted from Table 1 in revised manuscript.

  1. The integrity of science is an extremely important issue nowadays. The article serves this purpose when critically address the work of Gupta et al. (ref 100), but in my opinion this paragraph would need a bit of rephrasing.

Gupta et al [99] reported that the increased humoral response to the HERV-K dUTPase in psoriasis was probably due to cross reactivity with the Candida albicans encoded protein Ca019.10692. However, the 137 amino acid designated peptide they employed for their study, which was derived from the nucleocapsid domain of gag, exhibits no homology with the HERV-K dUTPase or with the PSORS1 dUTPase. Furthermore, the statement that C. albicans is known to be more frequent in psoriasis patients is illogical considering that HERVs encompass approximately 8% of the human genome. It is more likely that the increased antibody response observed against the HERV-K dUTPase is due to cross reactivity with the PSORS1 dUTPase.

Gupta et al used a peptide, named peptide-137, which is a 15 aa-long peptide

(KCYNCGQIGHLKKNC), that is truly from the gag protein and has no similarity to HERV-K

dUTPase. Please correct the text accordingly.I would also suggest that it should be explicitly said that their statement is not backed up by their experimental data. But there is no need to mention more about C. albicans (ie about its frequency) as that has no proven potential role in the increased humoral response.

We agree with the comment of the reviewer and the revised manuscript has been modified to state: “Gupta et al [97] reported that the increased humoral response to the HERV-K dUTPase in psoriasis was probably due to cross-reactivity with the Candida albicans encoded protein Ca019.10692. Notably, they employed a 15 amino acid long peptide (KCYNCGQIGHLKKNC) peptide,peptide-137) that has no similarity to the HERV-K dUTPase and their experimental data does not support this conclusion.”

  1. In addition to this in order to also serve integrity of science please note that in case of Ref 27 Tormo- Mas et al Mol. Cell 2013, the authors themselves and others revised the conclusions of Ref 27 (cf. PMID: 27112567, 25274731). The reference of this paper with outdated conclusions could be misleading, even the title of the paper contains a misconception, that dUTPases are similar to that of G-proteins, which has been proven to be untrue (cf. PMID: 27112567, 25274731). We agree with the reviewer and this reference has been removed from the revised manuscript.

  1. To further serve the purpose of science integrity please revise the sections about the BIV truncated dUTPases. It is rather an unfunctional dUTPase segment, than a validated dUTPase. It is impossible to validate it since it is truncated. It has been annotated as a dUTPase because it is between the RT and IN genes where DUT is found in FIV and EIAV. Although it admittedly share some sequence identity with the N-terminal of trimeric dUTPases it is not a dUTPase per se, rather just a segment of that. We agree with comment and the section on BIV has been removed from the revised manuscript.

  1. Table 2 ASVF dUTPase trimer but only 2 catalytic sites needs correction as the protein has 3 active sites, all built up by two adjacent chains (cf. PMID: 33139328, Ref 42 of the ms). This has been corrected in the revised manuscript.

Comments on the format of the manuscript:

  1. The text and tables would be more easy to follow if there would be cross-references between them. For example Figure 1 and Tables 1-3 should include the chapter number next to the name of viruses, and in case of tables the abbreviated name of the virus used in the text. This is an opinion of the reviewer. The Tables are inserted in the manuscript at the location of the relevant chapters so in our opinion it is not necessary to make additional changes.

  1. The authors describe the monomeric and trimeric dUTPases of viruses. I would suggest to have a figure with the 3D structure of these enzymes highlighting the motifs 1-5, and 6. This would aid the reader to follow and understand the text. This is a suggestion of both reviewers. However, we are not crystallographers and thus to obtain the data necessary to do this we would need to obtain permission from other journals. Furthermore, the pertinent material has been referenced.

  1. Figure 2 is rather a table now, I would suggest to include the multiple sequence alignment of these motifs on the figure, and I ask the authors to include the accession codes of the proteins and indicate the position of the residues involved in motif 6 within the full-length sequence. (Perhaps give the start and end positions). This latter is essential for the readers if they wish to have deeper look into this interesting topic. As suggested by the reviewer we have presented a multiple sequence alignment of Motif VI in Figure 2 and have presented accession codes and indicated the amino acids involved in this motif in the figure legend.

  1. Accession codes of proteins in Table 1-3 should be added. Genebank accession codes have been added to Tables 1-3 and Figure 2 in the revised manuscript.

  1. I think Chapter titles 2 and 3 should include the term dUTPase to emphasize that the chapters are about this protein in the viruses, not the viruses themselves. This has been done in the revised manuscript.

  1. From Line 336 and ref 99 the reference list is shifted as Gupta et al is not Ref 99 but Ref 100, Abergel et al is 126 not 125. Ref 99 should be included in the correct place and the reference numbers in text following Line 336 are to be refreshed. References have been corrected in revised manuscript.

  1. The style of references is not perfectly unified, sometimes doi-s appear, sometimes not, this should be fixed. Besides this formatting of Refs 53,57,66,117 should be slightly adjusted. Doi s appear for online journals which is consistent with the recommendations of Biomolecules.

  1. Star symbol in case of Table 3 is to be explained. In case of HHV6 it could be included in Table 3 that it has no enzymatic activity, and it can also contain that Pseudorabies virus dUTPase UL50 was proven to be enzymatically active. The Star symbol is actually in Table 2. It has been defined in the revised manuscript as follows: “* The beta herpesviruses lack motifs I-V and thus lack enzymatic activity. The UL72 and U45 have been describe as encoding putative dUTPases based upon sequence comparisons to alpha-and gamma-herpesviruses.”

  1. In case of WSSV dUTPase it could be included that dUTPase domain is at the N-terminal, while the C-terminal is of unknown function (as this is included in the text). This has been included in Table 2 in the revised manuscript.

Typos:

The expressions: in vivo and in vitro should be in italicized. This has been corrected.

Line 50 -helical dimeric proteins. This has been corrected.

Line 50 hydrolyzing both dUDP and dUTP. This has been corrected.

Table 1 Lizard-Cricket Iridiovirus variant IV (extra i) This has been corrected.

Line 194-195 While the myxovirus is replication-incompetent in humans, the use of replication competent vaccinia virus could express the poxvirus dUTPase in humans if used as

vaccine. This has been corrected.

Line 218 extra space 58 ] This has been corrected.

Line 221 as a fusion fusion protein This has been corrected.

Line 243 common motif order (I,II,III,IV and V) This has been corrected.

Line 325 HERV-K (not HERK-K). This has been corrected.

Line 377 type 1 (not type I to be consitent with other places at the text)  This has been corrected.

Line 475 extra space ORF11 This has been corrected.

Line 490 possibility mortality? This has been corrected.

Table 3 extra space [ 131,132]; [136-138, ] This has been corrected.

Reviewer 2 Report

This review article by Ariza and others summarizes novel functions of viral dUTPases in addition to its enzymatic activity. Although the authors are herpesvirus researchers, they provide comprehensive and succinct explanation on many other viruses, as well. This reviewer thinks this manuscript can be accepted only after minor modifications as follows.

This reviewer found some inappropriate expressions. Proofreading by a native scientific editor is recommended.

The first paragraph in the Introduction is almost a copy of the Abstract. Such repetitive sentences must be avoided.

The authors introduce 3 types of dUTPase in the introduction: homotrimeric, homodimeric, and monomeric. Schematic diagram of structures of the three types of the enzyme would help understanding of the readers.

Figure 2 here actually is not a figure, but a Table.

Fig2; the authors here simply list motif VI sequences of herpesvirus dUTPases. However, readers cannot tell if the sequences have similarity or not among them. So, this reviewer suggest that they present sequence alignment data as a figure, but not as a table (for reference, https://en.wikipedia.org/wiki/Sequence_alignment).

It is weird to find “4.5.3 alpha, beta, and gamma herpesvirinae” in the “4.5 gammaherpesvirinae”.

The reference numbers must be incorrect from some point. For example, [129] (line 468) must deal with LF1 and LF2, but it actually indicates a paper on gp120 in the Reference (line 821).

Author Response

Reviewer 2

  1. This reviewer found some inappropriate expressions. Proofreading by a native scientific editor is recommended. The reviewer indicated that  “English language and style are fine/minor spell check required”. While we agree the were some minor spelling/grammatical errors, we do not agree that a native scientific editor is required since we employed the use of spell/grammatical checker as well as the fact that three of the authors are native English speakers.
  2. The first paragraph in the Introduction is almost a copy of the Abstract. Such repetitive sentences must be avoided. We agree with the reviewer and this has been corrected in the revised manuscript by modifying the Introduction to read “While there are some exceptions, most free-living organisms encode for a deoxyuridine triphosphate nucleotidohydrolase (dUTPase; EC 3.6.1.23)[1]. In addition, many viruses which infect archaea, bacteria, plants and animals also encode for a dUTPase. Studies have demonstrated that the genes encoding for the dUTPases are essential in bacteria [2, 3], yeast [4] and mice [5], suggesting that this enzyme is required for survival/life. This concept is supported by the extensive number of species that possess dUTPases. Pfam, a database of protein families generated on annotations and sequence alignments using hidden Markov models have demonstrated the presence of putative dUTPases in at least 8358 species [6].”
  3. The authors introduce 3 types of dUTPase in the introduction: homotrimeric, homodimeric, and monomeric.Schematic diagram of structures of the three types of the enzyme would help understanding of the readers. This is a suggestion of both reviewers. However, we are not crystallographers and thus to obtain the data necessary to do this we would need to obtain permission from other journals. Furthermore, the pertinent material has been referenced. Also the homodimeric dUTPases as explained in the Introduction are not evolutionary related to the homotrimeric or monomeric dUTPases and thus are not a subject of this review manuscript. As discussed in comments to Reviewer 1 we are not crystallographers and thus to obtain the data necessary to do this we would need to obtain permission from other journals. Furthermore, the pertinent material has been referenced.
  4. Figure 2 here actually is not a figure, but a Table. Fig2; the authors here simply list motif VI sequences of herpesvirus dUTPases. However, readers cannot tell if the sequences have similarity or not among them. So, this reviewer suggest that they present sequence alignment data as a figure, but not as a table (for reference, https://en.wikipedia.org/wiki/Sequence_alignment). As suggested by the reviewer we have presented a multiple sequence alignment of Motif VI in Figure 2 and have presented accession codes and indicated the amino acids involved in this motif in the figure legend.

  1. It is weird to find “4.5.3 alpha, beta, and gamma herpesvirinae” in the “4.5 gammaherpesvirinae”. We agree and this was a mistake. It has been replaced in the revised manuscript to read 4.6.

  1. The reference numbers must be incorrect from some point. For example, [129] (line 468) must deal with LF1 and LF2, but it actually indicates a paper on gp120 in the Reference (line 821). We agree with this comment and the reference numbers have been corrected in the revised manuscript.

Round 2

Reviewer 1 Report

The authors made most of the requested corrections, and gave satisfactory answers, where changes were not made. The manuscript improved a lot so I recommend to publish it after some minor modifications listed below.

The text has not been corrected according to Comment 9. The manuscript still displays „Trimer but only has two catalytic sites” for ASFV in Table 2. It should rather state that "the protein has 3 active sites, all built up by two adjacent chains" (cf. PMID: 33139328, Ref 42 of the original ms).

I was also wondering why some of the dUTPases represented on Fig 2 were not included in the tables.

Line 35 space missing between „I,” and „II”

Line 56 Leishmania major „i” missing

Line 56 Trypanosoma „o” missing

Line 57 hydrolyzingboth dUDP and dUT (missing space before both and letter P from the end of the sentence ie: „dUDP and dUTP”

Line 142 accession code instead of „accession coded”

Line 144 missing space „aa263-297”

Line 145  NC_002686.2 instead of NC_002688.2

Line 148 NC-000898.1, instead of NC-00898.1, correct also in Table 2

Line 152 Murine gammaherpesvirus-68 (replace s with a)

Table 1 Pseudaletia (Mythimna) sp. granulovirus #8 accesion code is correctly „NC_033780.2” length of dUTPase is 145 (missing from the table)

Table 1 Equid gammaherpesvirus 2 NC_001350.1 is rather NC_001650.2,

Table 1 Yaba monkey tumor virus "Trimer" (not trimer)

Line 302 Alphaherpesvirinae (missing letter „l”)

Line 450 „class II Betaretroviruse-like” extra e

Line 491 „and thus their experimental data does not support this conclusion” (thus maybe added here)

It could also be (re-)included that the peptide „was derived from the nucleocapsid domain of gag”

Line 515 „molecules.These” missing space

Table 3 „UL83(tegument pp65)” space missing

Ref 27 „Convergent evolution involving” (missing letter „n” in convergent)

Ref 29 ” Derepression of SaPlbov1 is…” (missing „re” from derepression)

Ref 71 „immunogenenicity” is rather immunogenicity

Ref 1, Author name correctly is Grolmusz, not Grolimusz.

Author Response

Reply to Reviewer 1 comments. Our reply is indicated in bold italics.

The authors made most of the requested corrections, and gave satisfactory answers, where changes were not made. The manuscript improved a lot so I recommend to publish it after some minor modifications listed below.

  • The text has not been corrected according to Comment 9. The manuscript still displays „Trimer but only has two catalytic sites” for ASFV in Table 2. It should rather state that "the protein has 3 active sites, all built up by two adjacentchains" (cf. PMID: 33139328, Ref 42 of the original ms). This has been corrected in the revised manuscript.
  • I was also wondering why some of the dUTPases represented on Fig 2 were not included in the tables. The dUTPases were not included in Table 2 because there was no data available in the literature concerning the number of amino acids in the protein and the structure even though it can be assumed that the structures are monomeric.
  • Line 35 space missing between „I,” and „II” Corrected
  • Line 56 Leishmania major „i” missing Corrected
  • Line 56 Trypanosoma „o” missing Corrected
  • Line 57 hydrolyzingboth dUDP and dUT (missing space before both and letter P from the end of the sentence ie: „dUDP and dUTP” Corrected
  • Line 142 accession code instead of „accession coded” Corrected
  • Line 144 missing space „aa263-297” Corrected
  • Line 145 NC_002686.2 instead of NC_002688.2 Corrected
  • Line 148 NC-000898.1, instead of NC-00898.1, correct also in Table 2 Corrected
  • Line 152 Murine gammaherpesvirus-68 (replace s with a) Corrected
  • Table 1 Pseudaletia (Mythimna) sp. granulovirus #8 accesion code is correctly „NC_033780.2” length of dUTPase is 145 (missing from the table) Corrected
  • Table 1 Equid gammaherpesvirus 2 NC_001350.1 is rather NC_001650.2, Corrected
  • Table 1 Yaba monkey tumor virus "Trimer" (not trimer) Corrected
  • Line 302 Alphaherpesvirinae (missing letter „l”) Corrected
  • Line 450 „class II Betaretroviruse-like” extra e Corrected:actually line 470
  • Line 491 „and thus their experimental data does not support this conclusion” (thus maybe added here) Corrected
  • It could also be (re-)included that the peptide „was derived from the nucleocapsid domain of gag” This statement was re-incorporated.
  • Line 515 „molecules.These” missing space Corrected
  • Table 3 „UL83(tegument pp65)” space missing. Corrected
  • Ref 27 „Convergent evolution involving” (missing letter „n” in convergent) Corrected
  • Ref 29 ” Derepression of SaPlbov1 is…” (missing „re” from derepression) Corrected but actually reference 28 in revised manuscript
  • Ref 71 „immunogenenicity” is rather immunogenicity Corrected
  • Ref 1, Author name correctly is Grolmusz, not Grolimusz. Corrected
